# Learning to Reach Goals via Iterated Supervised Learning

**Dibya Ghosh**[*]
UC Berkeley

**Abhishek Gupta**[*]
UC Berkeley

**Ashwin Reddy**
UC Berkeley

**Justin Fu**
UC Berkeley

**Coline Devin**
UC Berkeley

**Benjamin Eysenbach**
Carnegie Mellon University

**Sergey Levine**
UC Berkeley

## Abstract

Current reinforcement learning (RL) algorithms can be brittle and difficult to use, especially when learning goal-reaching behaviors from sparse rewards. Although supervised imitation learning provides a simple and stable alternative, it requires access to demonstrations from a human supervisor. In this paper, we study RL algorithms that use imitation learning to acquire goal reaching policies from scratch, without the need for expert demonstrations or a value function. In lieu of demonstrations, we leverage the property that any trajectory is a successful demonstration for reaching the final state in that same trajectory. We propose a simple algorithm in which an agent continually relabels and imitates the trajectories it generates to progressively learn goal-reaching behaviors from scratch. Each iteration, the agent collects new trajectories using the latest policy, and maximizes the likelihood of the actions along these trajectories under the goal that was actually reached, so as to improve the policy. We formally show that this iterated supervised learning procedure optimizes a bound on the RL objective, derive performance bounds of the learned policy, and empirically demonstrate improved goal-reaching performance and robustness over current RL algorithms in several benchmark tasks.

## 1 Introduction

Reinforcement learning (RL) provides an elegant framework for agents to learn general-purpose behaviors supervised by only a reward signal. When combined with neural networks, RL has enabled many notable successes, but our most successful deep RL algorithms are far from a turnkey solution. Despite striving for data efficiency, RL algorithms, especially those using temporal difference learning, are highly sensitive to hyperparameters (Henderson et al., 2018) and face challenges of stability and optimization (Tsitsiklis & Van Roy, 1997; van Hasselt et al., 2018; Kumar et al., 2019b), making such algorithms difficult to use in practice.

If agents are supervised not with a reward signal, but rather demonstrations from an expert, the resulting class of algorithms is significantly more stable and easy to use. Imitation learning via behavioral cloning provides a simple paradigm for training control policies: maximizing the likelihood of optimal actions via supervised learning. Imitation learning algorithms using deep learning are mature and robust; these algorithms have demonstrated success in acquiring behaviors reliably from high-dimensional sensory data such as images (Bojarski et al., 2016; Lynch et al., 2019). Although imitation learning via supervised learning is not a replacement for RL – the paradigm is limited by the difficulty of obtaining kinesthetic demonstrations from a supervisor – the idea of learning policies via supervised learning can serve as inspiration for RL agents that learn behaviors from scratch.

In this paper, we present a simple RL algorithm for learning goal-directed policies that leverages the stability of supervised imitation learning without requiring an expert supervisor. We show that when learning goal-directed behaviors using RL, demonstrations of optimal behavior can be generated from sub-optimal data in a fully self-supervised manner using the principle of data relabeling: that every trajectory is a successful demonstration for the state that it *actually* reaches, even if it is sub-optimal

---

[*]First two authors contributed equally. Correspondence at `dibya.ghosh@berkeley.edu`

for the goal that was originally commanded to generate the trajectory. A similar observation of hindsight relabelling was originally made by Kaelbling (1993), more recently popularized in the deep RL literature (Andrychowicz et al., 2017), for learning with off-policy value-based methods and policy-gradient methods (Rauber et al., 2017). When goal-relabelling, these algorithms recompute the received rewards as though a different goal had been commanded. In this work, we instead notice that goal-relabelling to the final state in the trajectory allows an algorithm to re-interpret an action collected by a sub-optimal agent as though it were collected by an expert agent, just for a different goal. This leads to a substantially simpler algorithm that relies only on a supervised imitation learning primitive, avoiding the challenges of value function estimation. By generating demonstrations using hindsight relabelling, we are able to apply goal-conditioned imitation learning primitives (Gupta et al., 2019; Ding et al., 2019) on data collected by *sub-optimal* agents, not just from an expert supervisor.

We instantiate these ideas as an algorithm that we call goal-conditioned supervised learning (GCSL). At each iteration, trajectories are collected commanding the current goal-conditioned policy for some set of desired goals, and then relabeled using hindsight to be optimal for the set of goals that were actually reached. Supervised imitation learning with this generated "expert" data is used to train an improved goal-conditioned policy for the next iteration. Interestingly, this simple procedure provably optimizes a lower bound on a well-defined RL objective; by performing self-imitation on all of its own trajectories, an agent can iteratively improve its own policy to learn optimal goal-reaching behaviors without requiring any external demonstrations and without learning a value function. While self-imitation RL algorithms typically choose a small subset of trajectories to imitate (Oh et al., 2018; Hao et al., 2019) or learn a separate value function to reweight past experience (Neumann & Peters, 2009; Abdolmaleki et al., 2018; Peng et al., 2019), we show that GCSL learns efficiently while training on *every* previous trajectory without reweighting, thereby maximizing data reuse.

The main contribution of our work is GCSL, a *simple* goal-reaching RL algorithm that uses supervised learning to acquire policies from scratch. We show, both formally and empirically, that *any* trajectory taken by the agent can be turned into an optimal one using hindsight relabelling, and that imitation of these trajectories (provably) enables an agent to (iteratively) learn goal-reaching behaviors. That iteratively imitating all the data from a sub-optimal agent leads to optimal behavior is a non-trivial conclusion; we formally verify that the procedure optimizes a lower-bound on a goal-reaching RL objective and derive performance bounds when the supervised learning objective is sufficiently minimized. In practice, GCSL is simpler, more stable, and less sensitive to hyperparameters than value-based methods, while still retaining the benefits of off-policy learning. Moreover, GCSL can leverage demonstrations (if available) to accelerate learning. We demonstrate that GCSL outperforms value-based and policy gradient methods on several challenging robotic domains.

## 2 PRELIMINARIES

**Goal reaching.** The goal reaching problem is characterized by the tuple $\langle \mathcal{S}, \mathcal{A}, \mathcal{T}, \rho(s_0), T, p(g) \rangle$, where $\mathcal{S}$ and $\mathcal{A}$ are the state and action spaces, $\mathcal{T}(s'|s,a)$ is the transition function, $\rho(s_0)$ is the initial state distribution, $T$ the horizon length, and $p(g)$ is the distribution over goal states $g \in \mathcal{S}$. We aim to find a time-varying goal-conditioned policy $\pi(\cdot|s,g,h)$: $\mathcal{S} \times \mathcal{S} \times [T] \to \Delta(\mathcal{A})$, where $\Delta(\mathcal{A})$ is the probability simplex over the action space $\mathcal{A}$ and $h$ is the remaining horizon. We will say that a goal is achieved if the agent has reached the goal at the end of the episode. Correspondingly, the learning problem is to acquire a policy that maximizes the probability of achieving the desired goal:

$$J(\pi) = \mathbb{E}_{g \sim p(g)} \left[ P_{\pi_g} \left( s_T = g \right) \right]. \tag{1}$$

Notice that unlike a shortest-path objective, this final-timestep objective provides no incentive to find the shortest path to the goal. We shall see in Section 3 that this notion of optimality is more than a simple design choice: hindsight relabeling for optimality emerges naturally when maximizing the probability of achieving the goal, but does not when minimizing the time to reach the goal.

The final timestep objective is especially useful in practical applications where reaching a particular goal is challenging, but once a goal is reached, it is possible to remain at the goal. When reaching the goal is itself challenging, forcing the agent to reach the goal as fast as possible can make the learning problem unduly difficult. In contrast, this objective just requires the agent to eventually reach, and then stay at the goal, a more straightforward learning problem. In addition, the final timestep objective is useful when trying to learn robust solutions that potentially take longer over shorter solutions that

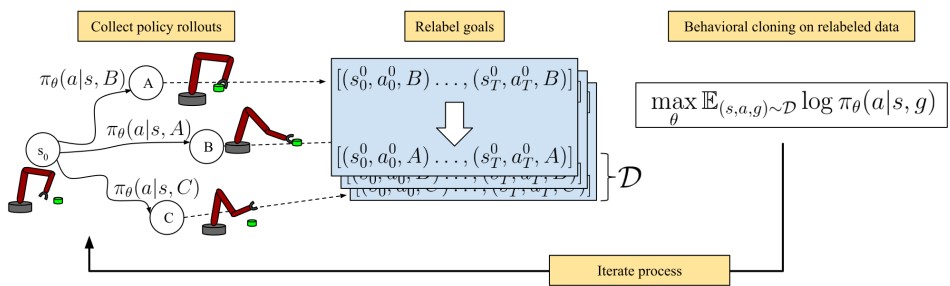

Figure 1: Goal-conditioned supervised learning (GCSL): The agent learns how to reach goals by sampling trajectories, relabeling the trajectories to be optimal in hindsight and treating them as expert data, and then performing supervised learning via behavioral cloning.

have a lower probability of success. There are certain drawbacks to this formulation: it is not as appropriate when the environment stochasticity prevents the agent from remaining at a goal, and it requires knowing an upper bound on the time it takes to reach a goal.

**Goal-conditioned RL.** The goal reaching problem can be equivalently defined using the nomenclature of RL as a collection of Markov decision processes (MDPs) $\{\mathcal{M}_g\}_{g \in \mathcal{S}}$. Each MDP $\mathcal{M}_g$ is defined as the tuple $\langle \mathcal{S}, \mathcal{A}, \mathcal{T}_g, r_g, \rho, T \rangle$, where the state space, action space, initial state distribution, and horizon as above. For each goal, a reward function is defined as $r_g(s) = \mathbb{1}(s = g)$. Using this notation, an optimal goal-conditioned policy maximizes the return in an MDP $\mathcal{M}_g$ sampled according to the goal distribution,

$$J(\pi) = \mathbb{E}_{g \sim p(g)} \left[ \mathbb{E}_{\tau \sim \pi_g} \left[ r_g(s_T) \right] \right]. \tag{2}$$

Since the transition dynamics are equivalent for different goals, off-policy value-based methods can use transitions collected for one goal to compute the value function for arbitrary other goals. Namely, Kaelbling (1993) first showed that if the transition $(s, a, s', r)$ was witnessed when reaching a specific goal $g$, it can be relabeled to $(s, a, s', r_{g'}(s))$ for an arbitrary goal $g' \in \mathcal{S}$ if the underlying goal reward function is known. Hindsight experience replay (Andrychowicz et al., 2017) considers a specific case of relabeling to when the relabeled goal is another state further down the trajectory.

**Goal-conditioned imitation learning.** If an agent is additionally provided expert demonstrations for reaching particular goals, behavioral cloning is a simple algorithm to learn the optimal policy by maximizing the likelihood of the demonstration data under the policy. Formally, demonstrations are provided as a dataset of expert behavior $\mathcal{D}^* = \{\tau_1, \tau_2, \dots\}$ from an expert policy $\pi^*$, where each trajectory $\tau_i = \{s_0^i, a_0^i, s_1^i, a_1^i, \dots s_T^i\}$ is optimal for reaching the final state in the trajectory. Given a parametric class of stochastic, time-varying policies $\Pi$, the behavioral cloning objective is to maximize the likelihood of actions seen in the data when attempting to reach this desired goal,

$$\pi_{BC} = \arg\max_{\pi \in \Pi} \mathbb{E}_{\tau \sim \pi^*} \left[ \log \pi(a_t | s = s_t, g = s_T, h = T - t) \right] \qquad \text{for } 0 \le t \le T.$$

## 3 Learning Goal-Conditioned Policies with Self-Imitation

In this section, we show how imitation learning via behavior cloning with data relabeling can be utilized in an iterative procedure that optimizes a lower bound on the RL objective. The resulting procedure, in which an agent continually relabels and imitates its own experience, is *not* an imitation learning algorithm, but rather an RL algorithm for learning goal-reaching from scratch without any expert demonstrations. This algorithm, illustrated in Fig. 1, is simple and allows us to perform off-policy reinforcement learning for goal reaching without learning value functions.

### 3.1 Goal-Conditioned Supervised Learning

We can attain the benefits of behavioral cloning without the dependence on human supervision by leveraging the following insight: under last-timestep optimality (Equation 1), a trajectory that fails to reach the intended goal is nonetheless optimal for reaching the goal it actually reached. As a result, a trajectory from a sub-optimal agent can be re-interpreted by goal-conditioned behavior cloning as an optimal trajectory for reaching a potentially different goal. This insight will allow us to convert sub-optimal trajectories into optimal goal reaching trajectories for different goals, without the need for any human supervision.

---

**Algorithm 1** Goal-Conditioned Supervised Learning (GCSL)

1: Initialize policy $\pi_1(\cdot \mid s, g, h)$
2: Initialize dataset $\mathcal{D}((s, a, g, h))$
3: **for** $k = 1, 2, 3, \ldots$ **do**
4:      Sample $g \sim p(g)$, collect data with $\pi_k(\cdot \mid \cdot, g)$.
5:      Log trajectory $\tau = (s_0, a_0, s_1, a_1, \ldots s_T, a_T)$
6:      Add tuples $\mathcal{D}_\tau$ to dataset $\mathcal{D}$                  $\triangleright$ see Eq. 3
7:      $\pi_{k+1} \leftarrow \arg\max_{\pi_\theta} \mathbb{E}_{\mathcal{D}} \left[\log \pi_\theta(a \mid s, g, h)\right]$
8: **end for**

---

More precisely, consider a trajectory $\tau = \{s_1, a_1, s_2, a_2, \ldots, s_T, a_T\}$ obtained by commanding the policy $\pi_\theta(a \mid s, g, h)$ to reach some goal $g$. For any time step $t$ and horizon $h$, the action $a_t$ in state $s_t$ is likely to be a good action for reaching $s_{t+h}$ in $h$ time steps (even if it is not a good action for reaching the originally commanded goal $g$), and thus can be treated as expert supervision for $\pi_\theta(\cdot \mid s_t, s_{t+h}, h)$. To obtain a concrete algorithm, we can relabel all time steps and horizons in a trajectory to create an expert dataset according to

$$\mathcal{D}_\tau = \{(s_t, a_t, g = s_{t+h}, h) : t, h > 0, t + h \leq T\}, \tag{3}$$

with states $s_t$, corresponding actions $a_t$, the corresponding goal set to future state $s_{t+h}$ and matching horizon $h$. Because the relabeling procedure is valid for any horizon, we can use any valid combination of $(s_t, a_t, s_{t+h}, h)$ tuples as supervision, for a total of $\binom{T}{2}$ optimal datapoints of $(s, a, g, h)$ from a single trajectory. This idea is related to data-relabeling for estimating the value function (Kaelbling, 1993; Andrychowicz et al., 2017; Rauber et al., 2017), but our work shows that data-relabelling can also be used to re-interpret data from a sub-optimal agent as though the data came from an optimal agent (with a different goal).

We then use this relabeled dataset for goal-conditioned behavior cloning. Algorithm 1 summarizes the approach: (1) Sample a goal from a target goal distribution $p(g)$. (2) Execute the current policy $\pi(a|s, g, h)$ for $T$ steps in the environment to collect a potentially suboptimal trajectory $\tau$. (3) Relabel the trajectory (Equation. 3) to add $\binom{T}{2}$ new expert tuples $(s_t, a_t, s_{t+h}, h)$ to the training dataset. (4) Perform supervised learning on the entire dataset to update the policy $\pi(a|s, g, h)$ via maximum likelihood. We term this iterative procedure of sampling trajectories, relabeling them, and training a policy until convergence *goal-conditioned supervised learning* (GCSL). This algorithm can use all of the prior off-policy data in the training dataset because this data continues to remain optimal under the notion of goal-reaching optimality that was defined in Section 2, but does not require any explicit value function learning.Perhaps surprisingly, this procedure optimizes a lower bound on an RL objective, as we will show in Section 3.2.

The GCSL algorithm (as described above) can learn to reach goals from the target distribution $p(g)$ simply using iterated behavioral cloning. This goal reaching algorithm is off-policy, optimizes a simple supervised learning objective, and is easy to implement and tune without the need for any explicit reward function engineering or demonstrations. Additionally, since GCSL uses a goal-conditioned imitation learning algorithm as a sub procedure, if demonstrations or off-policy data are available, it is easier to incorporate this data into training than with off-policy value function methods.

### 3.2 THEORETICAL ANALYSIS

We now formally analyze GCSL to verify that it solves the goal-reaching problem, quantify how errors in approximation of the objective manifest in goal-reaching performance, and understand how it relates to existing RL algorithms. Specifically, we derive the algorithm as the optimization of a lower bound of the true goal-reaching objective, and we show that under certain conditions on the environment, minimizing the GCSL objective enables performance guarantees on the learned policy.

We start by describing the objective function being optimized by GCSL. For ease of presentation, we make the simplifying assumption that the trajectories are collected from a single policy $\pi_{old}$, and that relabelling is only done with goals at the last timestep ($g = s_T$). GCSL performs goal-conditioned behavioral cloning on a distribution of trajectories $\pi_{old}(\tau) = \mathbb{E}_{g \sim p(g)}[\pi_{old}(\tau|g)]$, resulting in the following objective:

$$J_{\text{GCSL}}(\pi) = \mathbb{E}_{\tau \sim \pi_{old}(\tau)} \left[ \sum_{t=0}^{T} \log \pi(a = a_t | s = s_t, g = s_T, h = T - t) \right].$$

Our main result shows that, under certain assumptions about the off-policy data distribution, optimizing the GCSL objective $J_{\text{GCSL}}(\pi)$ optimizes a lower bound on the desired objective, $J(\pi)$.

**Theorem 3.1.** *Let $J_{GCSL}$ and $J$ be as defined above. Then,*

$$J(\pi) \geq J_{GCSL}(\pi) - 4T(T-1)\alpha^2 + C.$$

*Where $\alpha = \max_{s,g,h} D_{TV}(\pi(\cdot|s,g,h) \| \pi_{old}(\cdot|s,g,h))$ and C is a constant independent of $\pi$.*

The proof is in Appendix B.1. This theorem provides a lower-bound on the goal-reaching objective with equality for the optimal policy; akin to many proofs for direct policy search methods, the strongest guarantees are provided under on-policy data collection ($\alpha = 0$). The analysis raises two questions: can we quantify the tightness of the bound given by Theorem 3.1, and what does an optimal solution to the GCSL objective imply about performance on the true objective?

The tightness of the bound depends on two choices in the algorithm: how off-policy data from $\pi_{old}$ is used to optimize the objective, and how the relabeling step adjusts the exact distribution of data being trained on. We find that the looseness induced by the relabeling can be controlled by two factors: 1) the proportion of data that must be relabeled, and 2) the distance between the distribution of trajectories that needed to be relabeled and the distribution of trajectories that achieved the desired goal and were not relabeled. If either of these quantities is minimized to zero, the looseness of the bound that stems from relabeling also goes to zero. We present this analysis formally in Appendix B.2.

Even when data is collected from an off-policy distribution, optimizing the GCSL objective over the full state space can provide guarantees on the performance of the learned policy. We write $\pi^*$ to denote a policy that maximizes the true performance $J(\pi)$, and $\tilde{\pi}^*$ to denote the policy that maximizes the GCSL objective $J_{\text{GCSL}}(\pi)$ over the set of all policies. The following theorem provides such a performance guarantee for deterministic environments (proof in Appendix B.3):

**Theorem 3.2.** *Consider an environment with deterministic dynamics and a data-collection policy $\pi_{old}$ with full support. If $\max_{s,g,h} D_{TV}(\pi(a|s,g,h), \tilde{\pi}^*(a|s,g,h)) \leq \epsilon$, then $J(\pi^*) - J(\pi) < \epsilon T$.*

This theorem states that in an environment with deterministic transitions, the policy that maximizes the GCSL objective $J_{GCSL}(\pi)$ also maximizes the true performance $J(\pi)$. Furthermore, if the GCSL loss is approximately minimized, then performance guarantees can be given as a function of the error across the full state space. Whereas Theorem 3.1 shows that GCSL *always* optimizes a lower bound on the RL objective when iteratively re-collecting data with the updated policy, Theorem 3.2 shows that in certain environments, simply optimizing the GCSL objective from any off-policy data distribution without iterative data collection can also lead to convergence.

## 4 RELATED WORK

Our work studies the problem of goal-conditioned RL (Kaelbling, 1993) from sparse goal-reaching rewards. To maximize data-efficiency in the presence of sparse rewards, value function methods use off-policy hindsight relabeling methods such as hindsight experience replay (Andrychowicz et al., 2017) to relabel rewards and transitions retroactively (Schaul et al., 2015; Pong et al., 2018). Despite the potential for learning with hindsight, optimization of goal-conditioned value functions suffers from instability due to challenging critic estimation. Rauber et al. (2017) extends hindsight relabelling to policy gradient methods, but is hampered by high-variance importance weights that emerge from relabelling. Our method also relabels trajectories in hindsight, but does so in a completely different way: to supervise an imitation learning primitive to learn the optimal policy. Unlike these methods, GCSL does not maintain or estimate a value function, enabling a more stable learning problem, and more easily allowing the algorithm to incorporate off-policy data.

GCSL is inspired by supervised imitation learning (Billard et al., 2008; Hussein et al., 2017) via behavioral cloning (Pomerleau, 1989). Recent works have also considered imitation learning with goal relabeling for learning from human play data (Lynch et al., 2019; Gupta et al., 2019) or demonstrations (Ding et al., 2019). While GCSL is procedurally similar to Lynch et al. (2019) and Ding et al. (2019), it differs crucially on the type of data used to train the policy — GCSL is trained

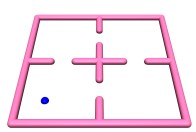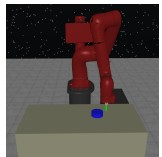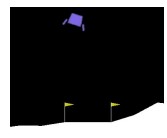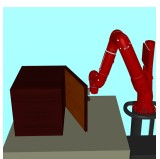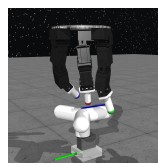

Figure 2: **Evaluation Tasks**: We study the following goal-reaching tasks: (from left to right) 2D navigation, robotic pushing, Lunar Lander, robotic door opening, dexterous object manipulation.

on data collected by the agent itself from *scratch*, not from an expert or (noisy) optimal supervisor. The fact that the same algorithmic procedure for training on optimal demonstrations can be applied iteratively using data from a sub-optimal agent to learn from scratch is non-trivial and constitutes one of our contributions.

GCSL has strong connections to direct policy search and self-imitation algorithms. Direct policy search methods (Mannor et al., 2003; Peters & Schaal, 2007; Theodorou et al., 2010; Goschin et al., 2013; Norouzi et al., 2016; Nachum et al., 2016) selectively weight policies or trajectories by their performance during learning, as measured by the environment's reward function or a learned value function, and maximize the likelihood of these trajectories using supervised learning. Similar algorithmic procedures have also been studied in the context of learning models for planning (Pathak et al., 2018; Savinov et al., 2018; Eysenbach et al., 2019). GCSL is also closely related to self-imitation learning, where a small subset of trajectories are chosen to be imitated alongside an RL objective (Oh et al., 2018; Hao et al., 2019), often measured using a well-shaped reward function. However, GCSL neither relies on a hand-shaped reward function nor chooses a select group of elites, instead using goal relabeling to imitate every previously collected trajectory for higher data re-use and sample efficiency. Goal-conditioned self-imitation learning methods when combined with a meta-controller, have been shown by Ecoffet et al. (2020) to learn policies much faster in a single-task sparse reward setting. This line of work presents an avenue for using GCSL to aid learning in the single-task setting as well.

## 5 EXPERIMENTAL EVALUATION

In our experiments, we comparatively evaluate GCSL on a number of goal-conditioned tasks. We focus on answering the following questions:

1. Does GCSL effectively learn goal-conditioned policies from scratch?
2. Can GCSL learn behaviors more effectively than standard RL methods?
3. Is GCSL less sensitive to hyperparameters than value-based methods?
4. Can GCSL incorporate demonstration data more effectively than value-based methods?

### 5.1 EXPERIMENTAL FRAMEWORK

We evaluate GCSL on five simulated control environments for goal-reaching: 2D room navigation, object pushing with a robotic arm, the classic Lunar Lander game, opening a door with a robotic arm, and object manipulation with a dexterous 9 DoF robotic hand (referred to as claw manipulation), shown in Figure 2 (Environments from Nair et al., 2018; Ghosh et al., 2019; Ahn et al., 2019, details in Appendix A.3). These tasks allow us to study the performance of our method under a variety of system dynamics, in settings with both easy and difficult exploration. For each task, the target goal distribution corresponds to a uniform distribution over reachable configurations. Performance is quantified by the distance of the agent to the goal at the last timestep. We present details about the environments, evaluation protocol, hyperparameters, and an extended set of results in Appendix A. We have additionally open-sourced our implementation at `https://github.com/dibyaghosh/gcsl`.

For the practical implementation of GCSL, we parameterize the policy as a neural network that takes in state, goal as input, ignoring the horizon, and outputs a distribution over actions. Although in general, the optimal policy does vary with the horizon, in environments where it is possible to stay at the goal, a Markovian policy that reaches, then stays at the goal can be near-optimal. Empirically, we also find that Markovian policies exhibit more coherent exploratory behavior than horizon-varying

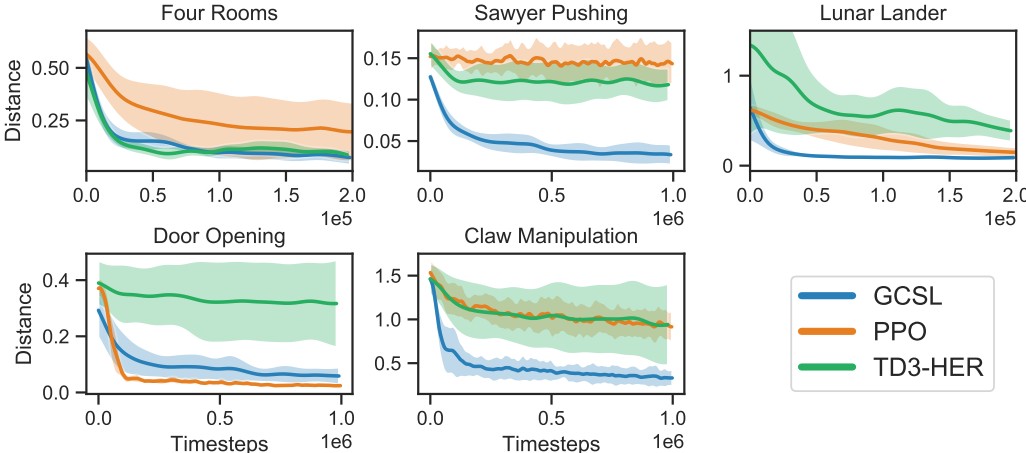

Figure 3: On a majority of tasks, GCSL performs well or better compared to more complex RL algorithms like PPO (Schulman et al., 2017) or TD3-HER (Andrychowicz et al., 2017). Shaded regions denote the standard deviation across 5 random seeds (lower is better).

policies. In our main results, the GCSL policy network ignores the horizon provided to the policy, a design decision we ablate in Section 5.3. Implementation details for GCSL are in Appendix A.1.

## 5.2 LEARNING GOAL-CONDITIONED POLICIES

We first evaluate the effectiveness of GCSL for reaching goals on the domains visualized in Figure 2, covering a variety of control problems spanning robotics and video games. To better understand the performance of our algorithm, we provide comparisons to value-based methods utilizing hindsight experience replay (HER) (Andrychowicz et al., 2017), and policy-gradient methods, two well established families of RL algorithms for solving goal-conditioned tasks. In particular, we compare against TD3-HER, an off-policy temporal difference RL algorithm that combines TD3 (Fujimoto et al., 2018) (an improvement on the DDPG method used by Andrychowicz et al. (2017)) with HER. TD3-HER requires significantly more machinery than GCSL: while GCSL only maintains a policy, TD3-HER maintains a policy, a value function, a target policy, and a target value function, all of which are necessary for good performance. We also compare with PPO (Schulman et al., 2017), a state-of-the-art on-policy policy gradient algorithm that does not leverage data relabeling, but is known to provide more stable optimization than off-policy methods and perform well on typical benchmark problems. Details for the training procedure for these comparisons, hyperparameter and architectural choices, as well as some additional comparisons are presented in Appendix A.2.

The results in Figure 3 show that GCSL generally performs as well or better than the best performing prior RL method on each task, only losing out slightly to PPO on the door opening task, where exploration is less of a challenge. GCSL outperforms both methods by a large margin on the pushing and claw tasks, and by a small margin on the lunar lander task. These empirical results suggest that GCSL, despite its simplicity, represents a stable and appealing alternative to significantly more complex RL methods, without the need for separate critics, policy gradients, or target networks.

## 5.3 ANALYSIS OF LEARNING PROGRESS AND LEARNED BEHAVIORS

To analyze GCSL, we evaluate its performance in a number of scenarios, varying the quality and quantity of data, the policy class, and the relabeling technique (Figure 5). Full details for these scenarios and results for all domains are in Appendix A.4.

First, we study how varying the policy class can affect the performance of GCSL. In Section 5.1, we hypothesized that GCSL with a Markovian policy would outperform a time-varying policy. Indeed, allowing policies to be time-varying ("Time-Varying Policy" in Figure 5) speeds up training on domains like Lunar Lander; on domains requiring more exploration like the Sawyer pushing task, exploration using time-varying policies is ineffective and degrades performance.

Figure 4: **Hyperparameter Robustness**: Distribution of final performance of GCSL and TD3-HER across nine hyperparameter configurations in each environment (see Section 5.4 for details). Higher values indicate better performance, and tightly clustered distributions indicate lower sensitivity to hyperparameters. GCSL is more performant and robust to hyperparameters than TD3-HER.

To investigate the impact of the data-collection policy, we consider variations that collect data using a fixed policy or train only on on-policy data. When collecting data using a fixed policy ("Fixed Data Collection" in Figure 5), the algorithm learns much slower, suggesting that iterative data collection is crucial for GCSL. By forcing the data to be on-policy ("On-Policy" in Figure 5), the algorithm cannot utilize all data seen during training. GCSL still makes progress in this case, but more slowly. We additionally consider limited-horizon relabeling, in which only states and goals that are at most 3 steps apart are relabeled, similar to proposals in prior work (Pathak et al., 2018; Savinov et al., 2018). Limiting the horizon degrades performance ("Limited relabeling" in Figure 5), indicating that multi-horizon relabeling is important.

Finally, we discuss the concern that since GCSL uses final-timestep optimality, it may provide significantly different behaviors than shortest-path optimality. While in theory, GCSL can learn round-about trajectories or otherwise exhibit pathological behavior, we find that on our empirical benchmarks, GCSL learns fairly direct goal-reaching behaviors (visualized in Appendix C.2). Since even the time-varying policy shares network parameters for different horizons, we hypothesize that the policy is constrained to produce behaviors that are roughly consistent through time, resulting in directed behaviors that resemble shortest-path optimality.

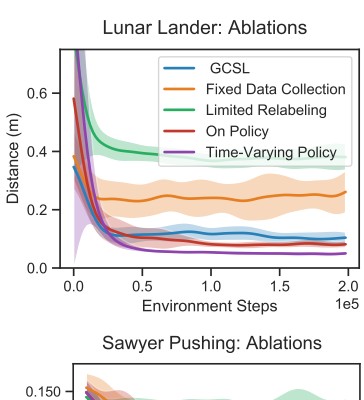

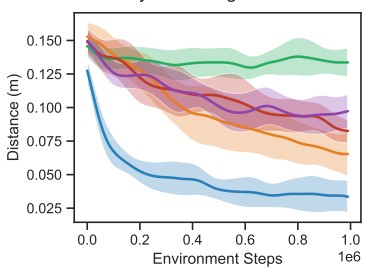

Figure 5: Ablations of GCSL on Lunar Lander and pushing. Other domains in Appendix A.4.

## 5.4 ROBUSTNESS TO HYPERPARAMETERS

Our next experiment tests the hypothesis that GCSL is more robust to hyperparameters than value-based RL methods like TD3-HER. The intuition is that, while dynamic programming methods are known to be quite sensitive to hyperparameters (Henderson et al., 2018), supervised learning techniques seem more robust. We ran a sweep across nine hyperparameter configurations, varying network capacity (size of the hidden layers in $[250, 500, 1000]$) and frequency of gradient updates (gradient updates per environment step in $[1, 2, 4]$). We compared both GCSL and TD3-HER and plotted the distribution of final timestep performance across all possible configurations in Fig. 4. We observe that the distribution of performance for GCSL is more tightly clustered than for TD3-HER, indicating lower sensitivity to hyperparameters. We emphasize that GCSL has fewer hyperparameters than TD3-HER; since GCSL does not learn a value function, it does not require parameters for the value function architecture, target update frequency, discount factor, or actor update frequency.

## 5.5 INITIALIZING WITH DEMONSTRATIONS

As GCSL can relabel and imitate trajectories from arbitrary sources, the algorithm is amenable to initialization from logs of previously collected trajectories or from demonstration data collected by

an expert. In this section, we compare the performance of GCSL bootstrapped from expert demonstrations to TD3-HER. Both methods can in principle utilize off-policy demonstrations; however, our results in Figure 6 show that GCSL benefits substantially more from these demonstrations. While value-based RL methods are known to struggle with data that is far off-policy (Kumar et al., 2019a), the simple supervised learning procedure in GCSL can take advantage of such data easily.

In this experiment, we provide the agent with a set of demonstration trajectories, each for reaching a different goal. GCSL adds this data to the initial dataset, without any other modifications to the algorithm. For TD3-HER, we incorporate demonstrations following the setup of Vecerik et al. (2017). Even with these measures, the value function in TD3-HER still suffers degraded performance and error accumulation during pre-training. When expert demonstrations are provided for the robotic pushing environment (Figure 6), GCSL progressively improves faster than when from scratch, but TD3 is unable to improve substantially beyond the original behavioral-cloned policy. We hypothesize that the difference in performance largely occurs because of the instability and optimism bias present when training value functions using demonstrations.

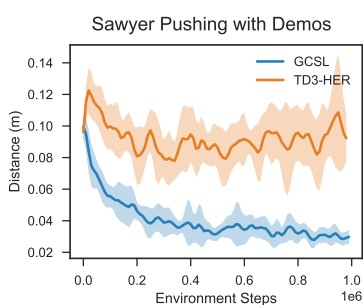

Figure 6: **Demonstrations:** GCSL incorporates expert demonstrations more effectively than TD3-HER.

## 6 Discussion

We proposed GCSL, a simple algorithm that uses supervised learning on its own previously collected data to iteratively learn goal-reaching policies from scratch. GCSL lifts several limitations of previous goal-reaching methods: it does not require a hand-defined reward, expert demonstrations, or the need to learn a value function. GCSL often outperforms more complex RL algorithms, is robust to hyperparameters, uses off-policy data, and can incorporate expert demonstrations when they are available. The current instantiation of GCSL is limited in exploration, since it relies primarily on the stochasticity of the policy to explore; a promising future direction would be to selectively reweight the sampled rollouts to promote novelty-seeking exploration. Nonetheless, GCSL is simple, scalable, and readily applicable — a step towards the fully autonomous learning of goal-directed agents.

## Acknowledgements

This research was supported by an NSF graduate fellowship, Berkeley DeepDrive, the National Science Foundation, the Office of Naval Research, and support from Google, Amazon, and NVIDIA. We thank Karol Hausman, Ignasi Clavera, Aviral Kumar, Marvin Zhang, Vikash Kumar for thoughtful discussions, insights and feedback on paper drafts.

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

# A   EXPERIMENTAL DETAILS

## A.1   GOAL-CONDITIONED SUPERVISED LEARNING (GCSL)

GCSL iteratively performs maximum likelihood estimation using a dataset of relabeled trajectories that have been previously collected by the agent. Here we present details about the policy class, data collection procedure, and other design choices.

We parameterize a time-invariant policy using a neural network which takes as input state and goal (not the horizon), and returns probabilities for a discretized grid of actions of the action space. The neural network concatenates the state and goal together, and passes the concatenated input into a feedforward network with two hidden layers of size 400 and 300 respectively, outputting logits for each discretized action. Empirically, we have found GCSL to perform much better with larger choices of neural networks; however, we use this two-layer neural network for fair comparisons to TD3-HER. The GCSL loss is optimized using the Adam optimizer with learning rate $\alpha = 5 \times 10^{-4}$, with a batch size of 256, taking one gradient step for every step in the environment.

When executing in the environment, the first 10000 environment steps are taken according to uniform random action selection, after which the data-collection policy is the greedy policy: $a = \arg\max_a \pi(a|s, g)$. The replay buffer stores trajectories and relabels on the fly, with the size of the buffer subject only to memory constraints. To clarify, instead of explicitly relabeling and storing all $\binom{T}{2}$ possible tuples from a trajectory, we instead save the trajectory and relabel at training time. When sampling from the dataset, a trajectory is chosen at random, a start index $t$ and goal index $t' > t$ are sampled uniformly at random, and the tuple corresponding to this state and goal are relabelled and sampled.

## A.2   RL COMPARISONS

We provide comparisons to two goal-reaching RL methods: TD3-HER and PPO. While GCSL is able to efficiently optimize the last-timestep objective, RL methods that learn value functions perform poorly when optimizing this objective. We include results of RL methods optimizing the last-timestep objective for completeness in Figure 9, but our main RL comparisons instead optimize the classical discounted sum of returns:

$$J(\pi) = \mathbb{E}_{g \sim p(g)}[\mathbb{E}_{\tau \sim \pi_g}[\sum_{t \leq T} \gamma^t r_g(s_t)]], \tag{4}$$

where reaching the goal leads to termination, the reward function remains the same, $r_g(s) = \mathbb{1}(s = g)$ and we use $\gamma = 0.99$. We log other metrics for these algorithms in Figure 12, and have included a variety of possible metrics.

**TD3-HER** (Fujimoto et al., 2018; Andrychowicz et al., 2017): To efficiently learn a value function, we use hindsight relabelling. Specifically, a transition $((s, g), a, (s', g))$ gets relabeled to $((s, g'), a, (s', g'))$, where $g' = g$ with probability 0.1, $g' = s'$ with probability 0.5, and $g' = s_t$ for some future state in the trajectory $s_t$ with probability 0.4. As described in Section 2, the agent receives a reward of 1 and the trajectory ends if the transition is relabeled to $g' = s'$, and 0 otherwise. Under this formalism, the optimal value function, $V^*(s, g) \propto \gamma^{T(s,g)}$, where $T(s, g)$ is the minimum expected time to go from $s$ to $g$. Both the Q-function and the actor for TD3 are parametrized as neural networks, with the same architecture (except final layers) for state-based domains as those for GCSL. We found the default values of learning rate, target update period, and number of critic updates to be the best amongst our hyperparameter search across the domains (single set of hyperparameters for all domains). Since GCSL uses discretized actions, we additionally compared to a version of TD3-HER that also uses discretized actions (results in Figure 12). The performance of TD3-HER w/ discretized actions decreases performance on the Claw task, increases on the Door Opening task, and has the same performance on the other three environments.

**PPO** (Schulman et al., 2017): Because PPO is an on-policy RL algorithm, we cannot relabel goals, unlike in GCSL or TD3-HER. Instead, we provide a surrogate $\epsilon$-ball indicator reward function: $r(s, g) = 1(d(s, g) < \epsilon)$, where $\epsilon$ is chosen appropriately for each environment. We emphasize that this reward function makes the policy optimization problem for PPO much easier, since while the other two methods only can check if two states are *exactly* equal, PPO also has access to the distance

(within an $\epsilon$-ball). To maximize the data efficiency of PPO, we performed a coarse hyperparameter sweep over the batch size for the algorithm. Just as with TD3, we mimic the same neural network architecture for the parametrizations of the policies as GCSL.

## A.3 TASK DESCRIPTIONS

For each environment, the goal space is identical to the state space; each trajectory in the environment lasts for 50 timesteps.

**2D Room Navigation** (Ghosh et al., 2019) This environment requires an agent to navigate to points in an environment with four rooms that connect to adjacent rooms. The state space has two dimensions, consisting of the cartesian coordinates of the agent. The agent has acceleration control, and the action space has two dimensions. The distribution of goals $p(g)$ is uniform on the state space, and the agent starts in a fixed location in the bottom left room.

**Robotic Pushing** (Ghosh et al., 2019) This environment requires a Sawyer manipulator to move a freely moving block in an enclosed play area with dimensions 40 cm $\times$ 20 cm. The state space is 4-dimensional, consisting of the Cartesian coordinates of the end-effector of the sawyer agent and the Cartesian coordinates of the block. The Sawyer is controlled via end-effector position control with a three-dimensional action space. The distribution of goals $p(g)$ is uniform on the state space (uniform block location and uniform end-effector location), and the agent starts with the block and end-effector both in the bottom-left corner of the play area.

**Lunar Lander** (Brockman et al., 2016) This environment requires a rocket to land in a specified region. The state space includes the normalized position of the rocket, the angle of the rocket, whether the legs of the rocket are touching the ground, and velocity information. Goals are sampled uniformly along the landing region, either touching the ground or hovering slightly above, with zero velocity.

**Door Opening:** (Nair et al., 2018) This environment requires a Sawyer manipulator to open a small cabinet door, initially shut closed, sitting on a table to a specified angle. The state space consists of the Cartesian coordinates of the Sawyer end-effector and the door's angle. As in the Robotic Pushing task, the three-dimensional action space controls the position of the end-effector. The distribution of goals $p(g)$ is uniform on door angles from 0 (completely closed) to 0.83 radians.

**Claw Manipulation:** (Ahn et al., 2019) A 9-DOF "claw"-like robot is required to turn a valve to various positions . The state space includes the positions of each joint of each claw (3 joints on 3 claws) and embeds the current angle of the valve in Cartesian coordinate ($\theta \mapsto (\sin\theta, \cos\theta)$). The robot is controlled via joint angle control. The goal space consists only of the claw angle, which is sampled uniformly from the unit circle.

## A.4 ABLATIONS

In Section 5.3, we analyzed the performance of the following variants of GCSL (Figure 7).

1. **Limited relabeling** - This model relabels only states and goals that are at most three steps apart: $\{(s_t, a_t, s_{t+h}, h) : t > 0, h \leq 3\}$

2. **On-Policy** Only the most recent 10000 transitions are stored and trained on.

3. **Fixed Data Collection** Data is collected according to a uniform policy over actions.

4. **Time-Varying Policy** Policies are conditioned on the remaining horizon. Alongside the state and goal, the policy gets a reverse temperature encoding of the horizon as input.

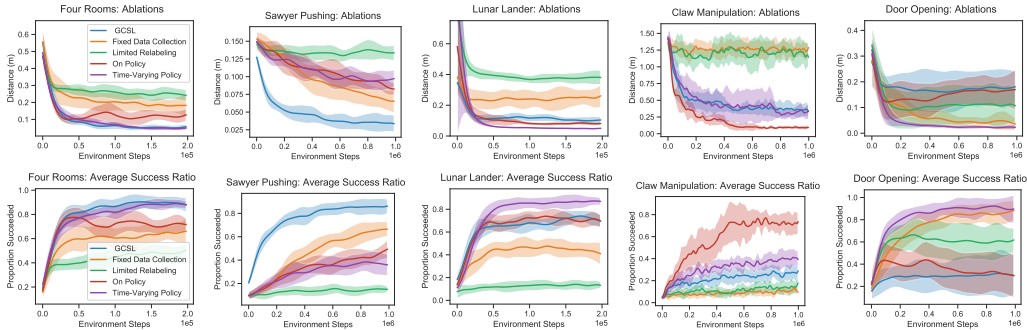

Figure 7: Performance across variations of GCSL (Section 5.3) for all experimental domains.

In addition, we also performed an ablation to measure how the performance of GCSL depends on the ratio of gradient steps to environment steps being collected. A larger number of gradient updates indicate higher levels of data re-use. Our results indicate that GCSL is robust to this hyperparameter.

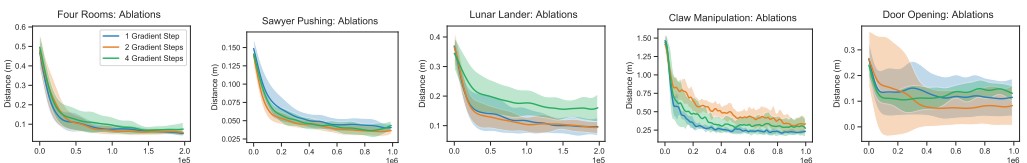

Figure 8: **Policy update frequency:** Performance when varying the ratio of policy update steps to environment steps. GCSL performs well even when significantly more gradient steps are taken on the replay buffer data.

Finally, we compared the performance of our RL comparisons, TD3-HER and PPO, when optimizing for the final-timestep objective in Equation 2 compared to optimizing the discounted return objective in Equation 4. To faithfully optimize the final-timestep objective, the policy and value networks for TD3-HER and PPO also take the remaining horizon as input. Our results indicate that TD3-HER and PPO learn slower (and sometimes not at all) when optimizing the final-timestep objective than the discounted return objective. Therefore, for the most fair evaluation, we compare the performance of GCSL to the RL methods that optimize the discounted return objective.

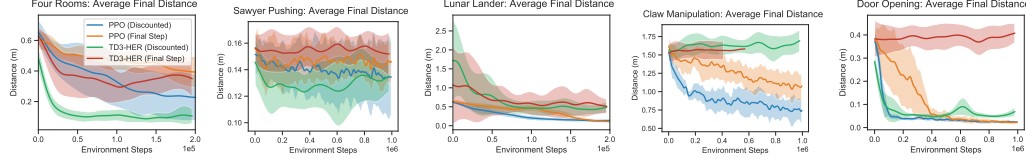

Figure 9: **Optimizing discounted return vs final-timestep objective**: Performance of TD3-HER and PPO when optimizing for the final time-step objective and the discounted return objective, as measured by median final distance to the goal. Both RL methods perform better with the discounted return objective uniformly across environments, so we use the discounted-return comparisons in the main paper.

## A.5    INITIALIZING WITH DEMONSTRATIONS

We train an expert policy for robotic pushing using TRPO with a shaped dense reward function, and collect a dataset of 200 trajectories, each corresponding to a different goal. To train GCSL using these demonstrations, we simply populate the replay buffer with these trajectories at the beginning of training, and optimize the GCSL objective using these trajectories to warm-start the algorithm. Initializing a value function method using demonstrates requires significantly more attention: we

perform the following procedure. First, we perform goal-conditioned behavior cloning to learn an initial policy $\pi_{BC}$. Next, we collect 200 new trajectories in the environment using a uniform data collection scheme. Using this dataset of 400 trajectories, we perform policy evaluation on $\pi_{BC}$ to learn $Q^{\pi_{BC}}$ using policy evaluation via bootstrapping. Having trained such an estimate of the Q-function, we initialize the policy and Q-function to these estimates, and run the appropriate value function RL algorithm.

## A.6   HINDSIGHT POLICY GRADIENTS

We compared GCSL to hindsight policy gradients (HPG) (Rauber et al., 2017) on the Fetch-Reach task from Andrychowicz et al. (2017). Whereas GCSL uses re-labelling in conjunction with supervised imitation learning, HPG performs reward relabeling with a policy gradient algorithm, introducing importance weights to correct for the distribution shift. As shown in Fig. 10, GCSL solves the task faster than HPG and converges to a policy that gets closer to the goal. HPG required more than 1 million samples to run reliably on Fetch-Reach, a task far easier to solve than any of our benchmark environments (Andrychowicz et al., 2017)

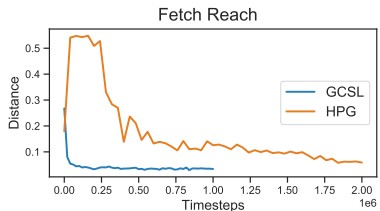

Figure 10:  GCSL converges faster and learns a more accurate goal-reaching policy than HPG (Rauber et al., 2017)

## A.7   SIMPLE EXPLORATION STRATEGIES

Since GCSL learns via self-imitation, like many other self-imitation methods, it faces challenges of poor exploration and converging to sub-optimal points (Oh et al., 2018). However, since GCSL can learn with off-policy data, we can deal with exploration challenges by using a directed exploration strategy or by injecting noise into the data collection process.

As a simple demonstration, we combined GCSL with an exploration strategy akin to Go-Explore (Ecoffet et al., 2020). This exploration strategy takes greedy actions with the policy for the first 90% of the trajectory, and then takes random actions for the remainder. Intuitively, this can be seen as a crude implementation of first getting to the fringe states, and then randomly exploring beyond the fringe. The main difference with this intuition is that we do not actually adjust the goal sampling strategy to only reach goals within the fringe. We tested this exploration strategy on a 2x larger version of the four rooms domain, where greedy exploration is insufficient to reach all regions

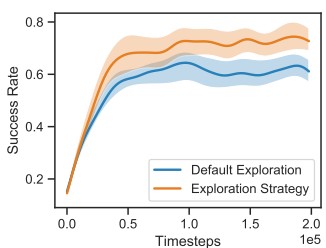

Figure 11: **Exploration strategies:** When combined with the exploration strategy from Ecoffet et al. (2020), GCSL explores more and learns a more accurate goal-reaching policy when navigating a 2x larger four-rooms domain.

in the space. GCSL with the default exploration strategy is unable to reach all the goals, but when combined with this new exploration strategy, is able to learn faster in this environment and reach 15% more goals.

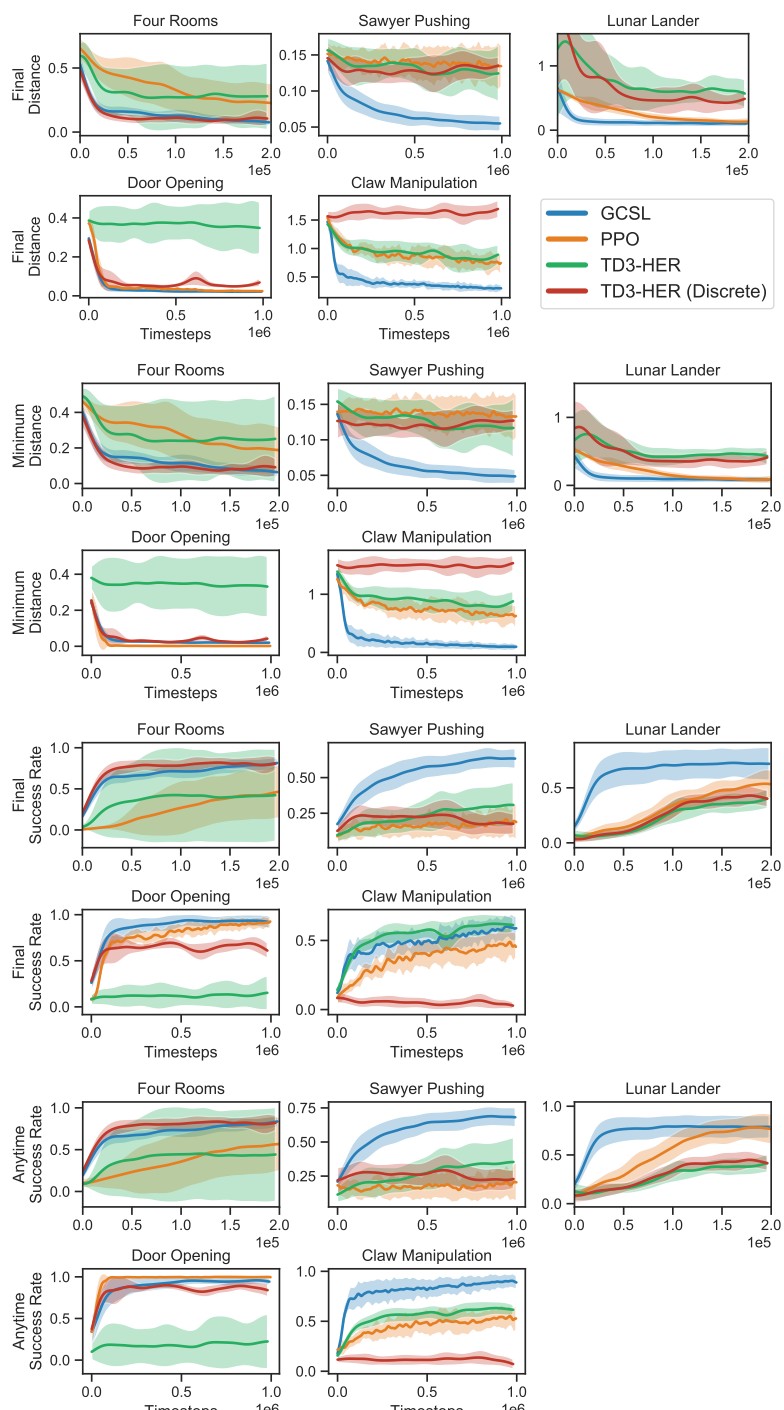

Figure 12: **Alternative Evaluation Metrics:** Here we present plots reporting (1) median distance to goal at the final timestep, (2) median minimum distance to goal within a trajectory (3) proportion of trajectories that were at the desired goal at the final timestep (3) proportion of trajectories that were at the desired goal at *any* timestep. As mentioned in Appendix A.2, we additionally provide comparisons to a version of TD3-HER that uses discretized actions. See Table 1 for time to goal metrics

## B  THEORETICAL ANALYSIS

### B.1  PROOF OF THEOREM 4.1

We will assume a discrete state space in this proof, and denote a trajectory as $\tau = \{s_0, a_0, \ldots, s_T, a_T\}$. Let the notation $\mathcal{G}(\tau) = s_T$ denote the final state of a trajectory, which represents the goal that the trajectory reached. As there can be multiple paths to a goal, we let $\tau_g = \{\tau : \mathcal{G}(\tau) = g\}$ denote the set of trajectories that reach a particular goal $g$. We abbreviate a policy's trajectory distribution as $\pi(\tau|g) = p(s_0) \prod_{t=0}^{T} \pi(a_t|s_t, g) \mathcal{T}(s_{t+1}|s_t, a_t)$. The target goal-reaching objective we wish to optimize is the probability of reaching a commanded goal, when goals are sampled from a pre-specified distribution $p(g)$.

$$J(\pi) = \mathbb{E}_{g \sim p(g), \tau \sim \pi(\tau|g)}[\mathbb{1}[\mathcal{G}(\tau) = g]]$$

GCSL optimizes the following objective, where the log-likelihood of the actions conditioned on the goals actually reached by the policy, $\mathcal{G}(\tau)$. The distribution of trajectories used to optimize the objective is collected through a different policy, $\pi_{old}$. We write $\pi_{old}(\tau) = \mathbb{E}_{g \sim p(g)}[\pi_{old}(\tau|g)]$ to concisely represent the marginalized distribution of trajectories from $\pi_{old}$.

$$J_{\text{GCSL}}(\pi) = \mathbb{E}_{\tau \sim \pi_{old}(\tau)}\left[\sum_{t=0}^{T} \log \pi(a_t|s_t, \mathcal{G}(\tau))\right]$$

To analyze how this objective relates to $J(\pi)$, we first analyze the relationship between $J(\pi)$ and a *surrogate objective*, given by

$$J_{\text{surr}}(\pi) = E_{g \sim p(g), \tau \sim \pi_{old}(\tau|g)}\left[\mathbb{1}[\mathcal{G}(\tau) = g] \log \pi(\tau|g)\right]$$

Theorem 1 from Schulman et al. (2015) states that

$$J(\pi) \geq J_{\text{surr}}(\pi) - \frac{4\gamma\epsilon}{(1-\gamma)^2}\alpha^2,$$

where $\gamma$ is a discount factor, $\epsilon$ is the maximum advantage over all states and actions, and $\alpha$ is the total variation distance between $\pi$ and $\pi_{old}$. It is straightforward to show that the bound can be rewritten in the finite-horizon undiscounted case in terms of the horizon $T$, following Kakade & Langford (2002); Ross et al. (2011), to obtain the bound

$$J(\pi) \geq J_{\text{surr}}(\pi) - 4T(T-1)\epsilon\alpha^2,$$

where $T$ is the horizon of the task. In the setting where data is collected from multiple policies, for example with a replay buffer, the bound cannot rely on the distance between policies at each state, but rather more generally the total variation distance between the trajectory distributions,

$$J(\pi) \geq J_{\text{surr}}(\pi) - \epsilon D_{TV}(\pi(\tau), \pi_{old}(\tau)). \tag{5}$$

Since our reward function is $\mathbb{1}[\mathcal{G}(\tau) = g]$, the return for any trajectory is bounded between $0$ and $1$, allowing us to bound $\epsilon$ above by $1$. This leaves $\alpha$, which is the total variation divergence between $\pi$ and $\pi_{old}$. This divergence may be high if the data collection policy is very far from the current policy, but is low if the data was collected via a recent policy.

We can now lower-bound the surrogate objective with the GCSL objective via the following:

$$
\begin{aligned}
J_{\text{surr}}(\pi) &= E_{g \sim p(g), \tau \sim \pi_{old}(\tau|g)}\left[\mathbb{1}[\mathcal{G}(\tau) = g] \log \pi(\tau|g)\right] \\
&= \sum_g p(g) \sum_\tau \pi_{old}(\tau|g) \log \pi(\tau|\mathcal{G}(\tau)) \mathbb{1}[\mathcal{G}(\tau) = g] \\
&= \sum_\tau \log \pi(\tau|\mathcal{G}(\tau)) \sum_g p(g) \pi_{old}(\tau|g) \mathbb{1}[\mathcal{G}(\tau) = g] \\
&= \sum_\tau \log \pi(\tau|\mathcal{G}(\tau)) \sum_g p(g) \pi_{old}(\tau|g) - \sum_\tau \log \pi(\tau|\mathcal{G}(\tau)) \sum_g p(g) \pi_{old}(\tau|g) \mathbb{1}[\mathcal{G}(\tau) \neq g]
\end{aligned}
\tag{6}
$$

$$
\begin{aligned}
&\geq \sum_\tau \log \pi(\tau|\mathcal{G}(\tau)) \sum_g p(g) \pi_{old}(\tau|g) \\
&= \mathbb{E}_{\tau \sim \mathbb{E}_g[\pi_{old}(\tau|g)]}[\log \pi(\tau|\mathcal{G}(\tau))].
\end{aligned}
$$

The final line is our goal-relabeling objective: we train the policy to reach goals we reached. The inequality holds since $\log \pi(\tau)$ is always negative. The inequality is loose by a term related to the probability of not reaching the commanded goal, which we analyze in the section below.

Since the initial state and transition probabilities do not depend on the policy, we can simplify $\log \pi(\tau|\mathcal{G}(\tau))$ as (by absorbing non $\pi$-dependent terms into $C_2$):

$$\mathbb{E}_{\tau \sim \pi_{old}(\tau)}[\log \pi(\tau|\mathcal{G}(\tau))] = \mathbb{E}_{\tau \sim \pi_{old}(\tau)} \left[ \log p(s_0) + \sum_{t=0}^{T} \log \pi(a_t|s_t, \mathcal{G}(\tau)) + \log \mathcal{T}(s_{t+1}|s_t, a_t) \right]$$

$$= \mathbb{E}_{\tau \sim \pi_{old}(\tau)]} \left[ \sum_{t=0}^{T} \log \pi(a_t|s_t\mathcal{G}(\tau)) \right] + C_2$$

$$= J_{\text{GCSL}}(\pi) + C_2.$$

Combining this result with the bound on the expected return completes the proof:

$$J(\pi) \geq J_{GCSL}(\pi) + C_1 + C_2 - 4T(T-1)\alpha^2$$

Note that in order for $J(\pi)$ and $J_{GCSL}(\pi)$ to be vacuously zero, the probability of reaching a goal under $\pi_{old}$ must be non-zero. This assumption is reasonable, and matches the assumptions on "exploratory data-collection" and full-support policies that are required by Q-learning and policy gradient convergence guarantees.

## B.2 QUANTIFYING THE QUALITY OF THE APPROXIMATION

The tightness of the bound presented above is controlled from two locations: the off-policyness of $\pi_{old}$ with respect to $\pi$ and the bound introduced by the lower bound in the theorem. The first is well-studied in policy gradient methods; in particular, when the data is on-policy, the gap between $J_{surr}(\pi)$ and $J(\pi)$ is known to be a policy-independent constant. We seek to better understand the gap introduced by Equation 6 in the analysis above.

We define $P_{\pi_{old}}(\mathcal{G}(\tau) \neq g)$ to be the probability of failure under $\pi_{old}$, and additionally define $p_{\text{wrong}}(\tau)$ and $p_{\text{right}}(\tau)$ to be the conditional distribution of trajectories under $\pi_{old}$ given that it did not reach and did the commanded goal respectively.

In the following section, we show that the gap introduced by Equation 6 can be controlled by the probability of making a mistake, $P_{\pi_{old}}(\mathcal{G}(\tau) \neq g)$, and $D_{TV}(p_{\text{wrong}}(\tau), p_{\text{right}}(\tau))$, a measure of the difference between the distribution of trajectories that must be relabeled and those not.

We rewrite Equation 6 as follows:

$$J_{surr}(\pi) = \sum_{\tau} \log \pi(\tau|\mathcal{G}(\tau)) \sum_{g} p(g)\pi_{old}(\tau|g) - \sum_{\tau} \log \pi(\tau|\mathcal{G}(\tau)) \sum_{g} p(g)\pi_{old}(\tau|g)\mathbb{1}[\mathcal{G}(\tau) \neq g]$$

$$= \mathbb{E}_{\tau \sim \pi_{old}}[\log \pi(\tau|\mathcal{G}(\tau))] - P_{\pi_{old}}(\mathcal{G}(\tau) \neq g)) \mathbb{E}_{\tau \sim p_{\text{wrong}}(\tau)}[\log \pi(\tau|\mathcal{G}(\tau))]$$

Define $D$ to be the Radon-Nikodym derivative of $p_{\text{wrong}}(\tau)$ wrt $\pi_{old}(\tau)$

$$= \mathbb{E}_{\tau \sim \pi_{old}(\tau)}[\log \pi(\tau|\mathcal{G}(\tau))] - P_{\pi_{old}}(\mathcal{G}(\tau) \neq g)) \mathbb{E}_{\tau \sim \pi_{old}(\tau)}[D \log \pi(\tau|\mathcal{G}(\tau))]$$

$$= (1 - P_{\pi_{old}}(\mathcal{G}(\tau) \neq g))\mathbb{E}_{\tau \sim \pi_{old}(\tau)}[\log \pi(\tau|\mathcal{G}(\tau))]$$

$$+ \underbrace{P_{\pi_{old}}(\mathcal{G}(\tau) \neq g)) \mathbb{E}_{\tau \sim \pi_{old}(\tau)}[(1 - D)\log \pi(\tau|\mathcal{G}(\tau))]}_{\text{Relevant Gap}}$$

The first term is affine with respect to the GCSL loss, so the second term is the error we seek to understand.

$$|\text{Relevant Gap}| = P_{\pi_{old}}(\mathcal{G}(\tau) \neq g) \left| \mathbb{E}_{\tau \sim \pi_{old}(\tau)}[(1 - D)\log \pi(\tau|\mathcal{G}(\tau))] \right|$$

$$\leq P_{\pi_{old}}(\mathcal{G}(\tau) \neq g)\mathbb{E}_{\tau \sim \pi_{old}}[|1 - D|]\mathbb{E}_{\tau \sim \pi_{old}(\tau)}[\log \pi(\tau|\mathcal{G}(\tau))]$$

$$= 2P_{\pi_{old}}(\mathcal{G}(\tau) \neq g)D_{TV}(\mathbb{E}_g[\pi_{old}(\tau|g)], p_{\text{wrong}}(\tau))\mathbb{E}_{\tau \sim \pi_{old}(\tau)}[\log \pi(\tau|\mathcal{G}(\tau))]$$

$$= 2P_{\pi_{old}}(\mathcal{G}(\tau) \neq g)(1 - P_{\pi_{old}}(\mathcal{G}(\tau) \neq g))D_{TV}(p_{\text{right}}(\tau), p_{\text{wrong}}(\tau))\mathbb{E}_{\tau \sim \pi_{old}(\tau)}[\log \pi(\tau|\mathcal{G}(\tau))]$$

The inequality is maintained because of the nonpositivity of $\log \pi(\tau)$, and the final step holds because $\pi_{old}(\tau)$ is a mixture of $p_{\text{wrong}}(\tau)$ and $p_{\text{right}}(\tau)$. This derivation shows that the gap between $J_{surr}$ and $J_{GCSL}$ (up to affine consideration) can be controlled by (1) the probability of reaching the wrong goal and (2) the divergence between the conditional distribution of trajectories which did reach the commanded goal (do not need to be relabeled) and those which did not reach the commanded goal (must be relabeled). As either term goes to $0$, this bound becomes tight.

### B.3 PROOF OF THEOREM 3.2

In this section, we now prove that sufficiently optimizing the GCSL objective over the full state space causes the probability of reaching the wrong goal to be bounded close to $0$, and thus bounds the gap close to $0$.

Suppose we collect trajectories from a policy $\pi_{old}$. Following the notation from before, we define $\pi_{old}(\tau) = \mathbb{E}_{g \sim p(g)}[\pi_{data}(\tau|g)]$. For convenience, we define $\pi^*(a_t|s_t, g) \propto \int_{\tau \setminus a_t} \pi_{data}(\tau)1(\mathcal{G}(\tau) = g)1(s_t(\tau) = s_t)$ to be the conditional distribution of actions for a given state given that the goal $g$ is reached at the end of the trajectory. If this conditional distribution is not defined, we let $\pi^*(a_t|s_t, g)$ be uniform, so that $\pi^*(a_t|s_t, g)$ is well-defined for all states, goals, and timesteps. The notation for $\pi^*$ is suggestive: in fact, it can be easily shown that under the assumptions of the theorem, full data coverage and deterministic dynamics, the induced policy $\pi^*$ is in fact the optimal policy for maximizing the probability of reaching the goal.

To show that the GCSL policy also incurs low error, we provide a coupling argument, similar to Schulman et al. (2015); Kakade & Langford (2002); Ross et al. (2011). Because $D_{TV}(\pi(a_t|s_t, g), \pi^*(a_t|s_t, g)) \leq \epsilon$, we can define a $(1 - \epsilon)$-coupled policy pair $(\overline{\pi}, \overline{\pi^*})$, which take differing actions with probability $\epsilon$. By a union bound over all timesteps, the probability that $\overline{\pi}$ and $\overline{\pi^*}$ take any different actions throughout the trajectory is bounded by $\epsilon T$, and because of the assumptions of deterministic dynamics, take the same trajectory with probability $1 - \epsilon T$. Now, since the two policies take different trajectories with probability *at most* $\epsilon T$, a simple bound shows that the probability that $\pi_{GCSL}$ reaches the goal is at most $\epsilon T$ less than $\pi*$, leading to our result that the performance gap $J(\pi^*) - J(\pi) < \epsilon T$. In environments in which every state is reachable from every other state in the desired horizon, this provides a global performance bound indicating that the optimal GCSL policy will reach the goal with probability at least $1 - \epsilon T$.

## C Directness of Policies Learned by GCSL

### C.1 Quantitative Analysis

We measure how direct the policies learned by GCSL are. To do so, we evaluate the following objective:

$$\text{ShortestTime}(\pi) = \mathbb{E}_{g \sim p(g)}[\mathbb{E}_{\tau \sim \pi_g}[\gamma^{T(\tau, g, \epsilon)}]] \quad T(\tau, g, \epsilon) = \inf\{t \in \mathbb{N} : d(s_t, g) < \epsilon\} \quad (7)$$

In other words, if the policy reaches the goal in $t$ timesteps, then the performance is $\gamma^t$, and if the agent does not, then the performance is $0$ ($t = \infty$). We choose this metric, as compared to exactly measuring the time taken to reach the goal, because it more cleanly handles the situation when the goal is not reached by the agent. We use $\gamma = 0.99$.

We additionally train an oracle agent using PPO to maximize a shaped reward $r(s, g) = -d(s, g)$ for 3 million timesteps. While the performance of this oracle agent and GCSL aren't directly comparable, since the oracle agent has privileged distance information, it serves as an rough upper bound on how fast an agent can reach the goal.

It may seem surprising that the GCSL policy learns to reach the goal so quickly, especially given that it optimizes the final-timestep optimality objective, and not the shortest-path objective. This is due to two factors. First, all the environments support the "reach the goal and stay at goal" behavior – if it were not possible to remain at the goal, then GCSL would not exhibit this type of behavior. Second, the policy we train ignores the horizon, and at a state, takes the same action at any timestep. This inductive bias on the policy prevents it from learning behaviors that remain still for a long period of time, and then go to the goal right before the time horizon is reached.

| Environment Name | Oracle | GCSL | PPO | TD3-HER |
|---|---|---|---|---|
| Four Rooms | 0.81 | 0.74 | 0.68 | 0.40 |
| Sawyer Pushing | 0.53 | 0.61 | 0.20 | 0.33 |
| Lunar Lander | 0.65 | 0.52 | 0.66 | 0.30 |
| Door Opening | 0.93 | 0.58 | 0.95 | 0.21 |
| Claw Manipulation | 0.70 | 0.68 | 0.45 | 0.55 |

Table 1: **Shortest-Time Objective (higher is better):** Values of $\text{ShortestTime}(\pi)$ for the final trained policy for each algorithm. $\text{ShortestTime}(\pi)$ is a smooth measure of how long it takes the agent to first reach the goal. GCSL reaches the goal nearly as quickly as the oracle policy on all the environments but the door opening task. The oracle expert policy is trained on a dense reward signal with PPO for significantly longer.

### C.2 Qualitative Analysis

Figure 13 below shows parts of the state along trajectories produced by GCSL. In Lunar Lander, this state is captured by the rocket's position, and in 2D Room Navigation it is the agent's position. While these trajectories do not always take the shortest path to the goal, they do often take fairly direct paths to the goal from the initial position avoiding very roundabout trajectories.

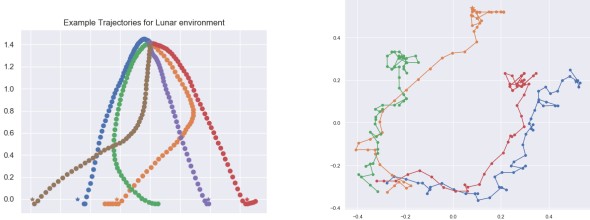

Figure 13: Examples of trajectories generated by GCSL for the Lunar Lander and 2D Room environments. Stars indicate the goal state.

