# OpenReview forum: "Learning to Reach Goals via Iterated Supervised Learning"
_ICLR.cc/2021/Conference — ICLR 2021 Oral_

### Official Review · AnonReviewer2 · 2020-10-28
**Official Blind Review #2**

**Rating:** 8
**Confidence:** 4

**Review:**

### Summary

The authors propose “GCSL” (goal-conditioned supervised learning), an algorithm that bridges the gap between reinforcement learning and imitation learning. Specifically, the authors are motivated by the limitations that RL is brittle when used with sparse rewards, and that IL requires expert demonstrations. Their technique performs learning without expert demonstrations, and does not require a learned value function or reward function. The authors formulate GCSL as iterative trajectory collection, goal relabeling, and policy refinement through behavioral cloning, and formally derive performance bounds on this technique. In addition, the authors demonstrate strong goal-reaching performance and robustness improvements over current RL techniques.

### Strengths

- The authors’ proposal is strongly motivated, in that RL techniques are clearly sensitive to hyperparameters and face stability challenges, and that using demonstrations is more robust. The paper is clearly written and the authors very cleanly explain their key ideas and insights.
- The authors’ proposed algorithm is based on a simple idea, in a very good way. Their technique is stable, requires no value functions or reward function engineering (as is common with traditional RL techniques), and their technique of relabeling generates a large number of samples, resulting in better data efficiency.
- The authors’ main theoretical insight that iterating on data from sub-optimal agents leads to optimal behavior is both non-trivial as the authors claim, and can likely be used to inspire other similar ideas or areas of study.
- The authors clearly showcase the experimental use cases of their technique by demonstrating its benefits in terms of stability to hyperparameters and in leveraging expert demonstrations. Apart from this strong set of experimental results, the authors also present detailed ablations in the supplementary pages that are very convincing.

---

### Weaknesses

- The theory section does seem a bit contrived. Specifically, optimality emerges when using the objective proposed by the authors, even if this is not commonly used. Further, the proof assumes that trajectories are collected from a single policy and relabeling is only performed on the last timestep, whereas in practice these conditions are not met. The performance guarantee also only holds with deterministic transitions. Finally, the guarantee on the convergence behavior is a good property, but is maybe less meaningful if in practice GCSL is difficult to converge to (if the optimization is challenging).
- Notably, the authors do not evaluate the generalization ability of their technique, and instead evaluate with the same train and test environment. Although this is common in RL, it would be interesting to see how GCSL performs in environments that require more sample efficiency and test generalization, like Procgen.
- The authors quantify performance as the distance of the agent to the goal at the last timestep. I’m not convinced this evaluation metric makes the most sense, especially when success ratio is the main metric we care about.
- On door opening, do the authors have an idea for why TD3+HER performs so much worse than PPO or GCSL? Similarly, on Sawyer pushing, do the authors have a hypothesis for why PPO and TD3+HER do not learn at all? Is there potentially a qualitative analysis of this unexpected behavior that can be performed? What about GCSL makes it so much more successful on Sawyer pushing?
- I’m surprised that GCSL is not any more sample efficient, since PPO and broadly speaking other policy gradient techniques are generally well-understood to be greedy sample-wise.
- The authors earlier in the paper hypothesized that an optimal policy would be non-Markovian, but that GCSL with a Markovian policy would outperform a time-varying one. This seems rather counter-intuitive to me, why do the authors suspect this is the case? Is the model overly exploiting instead of exploring when conditioned on the remaining horizon? Does this behavior change depending on the number of timesteps in an episode?
- In Figure 4, do the on-policy methods that converge slower do so with an empirically visible better convergence guarantee? For instance, even if it takes longer, the technique is guaranteed in the long-run to arrive at an optima?
- In Figure 4, the authors compare against limited-horizon relabeling as in prior work and show this method has drawbacks. Are the drawbacks due to the lack of multi-horizon relabeling, or due to having much fewer trajectories after relabeling?
- The authors compare robustness to hyperparameters against TD3, but do not compare against PPO. Because PPO is generally well-understood to not require much hyperparameter tuning empirically, I would recommend including this comparison as well.
- The authors claim qualitatively (in Appendix C) that despite the different objective, the learned trajectories generally take a direct path to the goal. Is there a way to quantify this quantitatively against an oracle that does take the shortest path; for example by comparing the time taken to get to the goal, or the time spent at the goal waiting for the episode to complete?

---

### Recommendation

Overall, I vote for accepting. I think this current submission is already relatively convincing as an accept, as it is clearly written, has well explained motivations, strong experimental results, and extensive ablations in the supplementary pages. The authors’ idea is conceptually simple, in a good way. My main gripe is that the theoretical results are only weakly held, or less relevant in practical settings, but that is rather typical and the strong experimental results do speak for themselves. I do have a few clarifying questions on the experimental results, but am regardless confident that this paper meets the ICLR acceptance criteria.

---

> ### Author Response · Authors · 2020-11-13
> **Response to Reviewer 2**
>
> Thank you for your valuable feedback!
>
> _Theory:_ We agree that there is a gap between the theoretical results and the practical settings evaluated in this paper. The intent of our analysis was to illustrate the type of formal guarantees that can be realized with algorithms like GCSL. For clarity, our theoretical analysis focuses on a simplified setting, last-timestep relabelling and single data-collection policy, but these assumptions can be lifted (with admittedly weaker guarantees). We have updated Appendix B (around Eq 5) to discuss an alternate form of the lower bound in the replay buffer setting, when the single data-collection policy assumption is removed.
>
> _Generalization:_ We are also very interested in understanding how the generalization of agents learned with supervised learning (like GCSL) compares to those learned using RL. However, we believe that properly investigating this research direction would be out of the scope of this particular manuscript.
>
> **Empirical Clarifications**
>
> _Distance vs success on final timestep:_ **Figure 12** in Appendix A (pg 16) shows learning curves for both distance to goal and success. We have now also added two new metrics to better understand the behaviors: “success at any timestep” and “minimum distance to goal within a trajectory”.
>
> _Quantitative analysis of “direct” paths:_ We have updated the manuscript to provide a measure of how fast the GCSL policy reaches the goal in **Appendix C.1, Table 1 (pg 20)**. On all domains but the Door task, GCSL reaches the goal about as fast as an RL agent optimized to find the shortest path to goal. On the Door task, the GCSL policy reaches the goal more slowly.
>
> _PPO hyperparameter sweep:_ We are currently running a robustness sweep of PPO, adjusting network architecture and batch size. As this is a compute-intensive experiment, we will update the manuscript with these results when they are completed.
>
> _Qualitative evaluation on Pusher and Door:_ On the pushing environment, the policies learned by PPO and TD3-HER can push the puck to a subset of goals (the easiest region), but do not reliably reach all the goals in the environment. We believe that this is because PG and value-based methods often face greater challenges in multi-task learning than on supervised learning objectives (see for example [Teh 17,  Schaul19]). On the Door task, the TD3 agent learns a degenerate policy that always pulls the door open: this skews the distribution of replay buffer data that is used for relabelling and training to be concentrated in a single region.
>
> _Non-markovian policies:_ We hypothesize that the Markovian policy performs better here because during data collection, it produces more directed exploration behaviors, as compared to the time-varying policy. As you mention also, the Markovian policy is likely to be more robust to potential misspecification of the horizon. We have updated the discussion in **Section 5.1** to reflect these points.
>
> _On-policy variant:_ Although the on-policy variant does have convergence guarantees in the tabular setting, in our benchmark tasks, the on-policy variant (even when run for longer) did not demonstrate better final performance than when the full replay buffer. Our hypothesis for this observation is that when keeping off-policy data, GCSL trains on a more diverse distribution of states and goals, which compensates for the data not being fully on-policy.
>
> _Multi-horizon relabelling:_ The drawbacks are due to the lack of multi-horizon relabelling. Increasing the number of trajectories to relabel (by running for longer) does not prevent the reduction in final performance.
>
>
> [1] YW Teh et. al. Distral: Robust multitask reinforcement learning
> [2] T Schaul, D Borsa, J Modayil, R Pascanu. Ray Interference: a Source of Plateaus in Deep Reinforcement Learning

---

### Official Review · AnonReviewer4 · 2020-10-28

**Rating:** 7
**Confidence:** 4

**Review:**

The paper proposes a new RL algorithms dedicated to learning goal-oriented policies. In the described setting, a policy has to reach a particular goal in T steps such that s_T = goal i.e the objective is not to reach the goal as soon as possible but in a particular number of steps. The proposed algorithm is very simple: it is an iterative algorithm where, at each iteration, i) a policy is used to collect trajectories. ii) Then, trajectories are relabelled based on the reached states -- reached states are becoming goals in the relabelling, iii) a new policy is learned by using behavioral cloning. In addition to the algorithm, the authors provide some theoretical insight about how and in which conditions one can prove that the algorithm is working well. At last, experiments are made on different environments and compared to some baselines showing that it allows to solve complex problems in less learning iterations. Moreover, an ablation study is provided to allow one to understand the effect of the different aspects of the learning technique.

First, the proposed technique is very simple, sharing some similarities with existing techniques (hindsight relabelling), but simplifying these techniques by just using behavioral cloning to discover a new policy instead of other and more complex value-based techniques. This idea may have two advantages: i) it is simple and ii) it may benefit of the robustness of supervised learning techniques, and particularly the fact they supervised learning may need less hyperparameters than other RL approaches. This is an interesting point and a good aspect of the article.

But the consequence seems to be that the algorithm is optimizing a criterion which is not the classical criterion used when learning goal-oriented policies. Indeed, here, the objective is to find a policy that reach a particular goal in T steps, and not to reach a goal as soon as possible. I think that this aspect is opening different questions:

A) the first one is the interest of learning such policies. The article does not really justify why and in which applications discovering such policies may be interesting. Indeed, reaching a goal in T steps may appear more difficult than reaching a particular goal in less steps, or as fast as possible. Moreover, in a stochastic environment, reaching a goal in T steps may be impossible due to the stochasticity of the environment while reaching the goal in less than T steps may be easier. I think that a better discussion on that point would improve the quality of the paper, while right now, it is not clear to me when such an objective function is interesting to study.

B) The second problem is related and in the experimental section, when comparing to classical RL techniques. Indeed, the reward function defined in this paper is very specific (i.e the agent get one if it reaches the goal at the last state of the trajectory), and classical RL techniques are using a discount factor lower than one which thus encourages the models to discover policies reaching goals as soon as possible. Again, a discussion explaining how exactly the comparison is made, and what is exactly compared would be interesting. It gives me the impression that RL techniques are used in a particularly difficult setting, why they may perform well on different but interesting reward functions encouraging for instance to reach the goal as soon as possible, or to reach the goal once during the T timesteps (and not only at the last state).

C) Also, some aspects are confusing: in section 5.1, it is written "we parameterize the policy as a neural network that takes in state, goal, and horizon as input" while in appendix A.1, it is written "We parameterize a time-invariant policy using a neural network which takes as input state and goal (not the horizon)," which is completely different. This is a crucial aspect since I do not understand how a policy that is not using the horizon as input is able to reach a goal at a particular timestep.

Other  comments:
Theorem 3.2 focused on deterministic environments is not really interesting since it considers a very particular case which is not a classical RL setting.
The way T is chosen is not clear to me. For instance, in many environments, in order to be able to reach interesting goals, T may have to be very large, and is unknown until a good policy is  known.
The extension about using demonstrations is an interesting aspect that would be interesting to develop more in the paper

Summary:
 The paper proposes an interesting and simple model that optimizes a particular and not classical objective. It is not clear if this objective is really interesting, and in which applications such a method can be used. The comparison with other techniques is fair, but considering this very particular objective while other objectives may be easier to learn and more interesting. Some aspects are not clear in the paper and the experimental section has to be improved.


Regarding the answers provided by the authors, I increase my score

---

> ### Author Response · Authors · 2020-11-13
> **Response to Reviewer 4**
>
>
> Thank you for your valuable feedback!
>
> To address your concerns about the consequences of using the final-timestep optimality objective, we have updated the manuscript to (details below)
>
> _A.1)_ add a paragraph to Section 2 (Goal Reaching) that discusses holistic situations where final-timestep optimality can be useful
> _A.2)_ add benchmarks of learned GCSL policies under more “conventional” objectives  (Appendix A Figure 12) and (Appendix C Table 1), and show that the final-timestep objective can serve as a useful surrogate for other goal-reaching objectives
> _B)_ clarify how the RL comparison objectives were trained and evaluated (Appendix A.2).
> _C)_ clarify how the GCSL policy is parameterized (in Section 5.1)
>
>
> The main reason we use the final-timestep objective is that it yields a simple and elegant learning algorithm. In some cases, the final-timestep objective is the most appropriate objective (A.1), and in other problems, less so. Even when it is not, empirically we find that it is often close enough to serve as a useful surrogate for many other objectives of interest (A.2).
>
> **A.1) The “final-timestep” objective in practical situations:**  The final timestep objective is a natural way to codify many robotic manipulation tasks; for example, a household robot may not be required to set a dinner table as soon as possible, but rather just reach this end configuration accurately by dinner time. In general, when reaching the goal at all is itself challenging, forcing the agent to reach the goal as fast as possible can make the learning problem unduly difficult, whereas the final timestep objective just requires the agent to eventually reach and then stay at the goal. We do acknowledge, as you pointed out, the final timestep objective is not appropriate in all settings: for example, when the environment stochasticity prevents the agent from remaining at a goal, or when we do not know an upper bound on the time it takes to reach a goal. However, it is applicable in many settings of practical interest. We have added a paragraph that paraphrases this discussion in **Section 2 (Goal Reaching).**
>
> **A.2) The final-timestep objective as a surrogate:** Even when the desired objective is not the final time-step optimality, we empirically find that optimizing this objective with GCSL can serve as a useful surrogate. During the rebuttal, we evaluated the performance of GCSL with alternative evaluation metrics like “reaching the goal within T timesteps” and “time to reach goal”, and found that it performs well under these criteria as well. Results with these alternate metrics are described and shown in **(Appendix A Figure 12)** and **(Appendix C Table 1)**. We hypothesize that GCSL works well under these metrics because on tasks where it is possible to do so, the behavior of GCSL policies can be qualitatively described as “reaches goal and stays at goal” (visualized in Appendix C.2).
>
> **B) RL Comparisons:**  We apologize for the confusion: in fact, the TD3-HER and PPO baselines reported in the paper are trained to optimize the discounted return:
> $\sum_{t \leq T} \gamma^T1(s=g)$
> We made this decision, because as you mention, the versions of TD3-HER and PPO that do faithfully optimize the final-timestep objective (as described by Eq 1) learn slower, likely because it poses a challenging value estimation problem.
>
> For completeness, we compared RL methods trained using the discounted return objective versus those trained with the final-timestep objective in **Appendix A.4 Figure 9.** These results indicate that the RL policies trained using the discounted reward in fact outperform those with the final-timestep objective, even when evaluated in terms of success at the final timestep.
>
> **C) Markovian vs Time-Conditioned Policies:** In our main experiments, GCSL uses a Markovian policy network that doesn’t take horizon as input. While it may seem surprising that a policy that doesn’t use horizon as input can reach a goal at a particular time-step, this still enables an agent to learn behaviors that reaches the goal directly and stays at the goal. As you mention, this may not work for all environments, especially those where it is impossible to remain at the goal location. For these environments, using a horizon-varying policy is more appropriate. We have updated **Section 5.1** to include this discussion, and to clarify the contradiction between Section 5.1 and Appendix A.1.
>
> We ablated our design decision to use a Markovian policy, and compared to a time-conditioned policy network in Section 5.3 (Figure 4 “time-varying policy”). The time-varying policy achieves similar levels of final performance on all the tasks, but learns slower than the Markovian policy, most notably on the Pusher. We hypothesize this to be an issue of exploration: a sub-optimal time-conditioned policy may choose very different actions at different timesteps, and not produce coordinated exploratory behaviors.

---

### Official Review · AnonReviewer1 · 2020-10-28
**Nice paper with an interesting algorithm, which could be made better by more discussion on some related work and potential limitations.**

**Rating:** 8
**Confidence:** 4

**Review:**

In the paper "Learning to Reach Goals via Iterated Supervised Learning", the authors propose a new approach to build conditional policies for reaching tasks that can reuse the previous failed attempts as new examples on how to reach the state that was actually reached during the failed execution. This approach is similar to the approach introduced in HER, but is based on Behavioural Cloning algorithm (thus the name self-imitation) instead of a value function.

The paper is overall well written, clearly illustrated and appropriately structured.

In my opinion, the proposed method shares significant similarities with the "First return then explore" paper (Ecoffet et al. 2020), which is a new version of the GO-Explore algorithm that builds a similar conditioned policy to "return" to a state and then explore from this state using a random action, which is then used to extend and improve the learned policy. Given the strong links between these two papers, I am surprised to not see this discussed related work section and compared in the experimental section.

My main concern about the algorithm is the risk of a lack of exploration. An extreme example is a degenerated policy that outputs 0 (i.e., no movement) regardless of its inputs. The relabelling will just reinforce this behaviour and the algorithms will quickly converge in a local optimum (which is actually far from being optimal). Obviously, this is an extreme example which is quite unlikely, but the same can happen if some states/goals are very unlikely to be observed given the policy. This can happen for instance, if the policy has a strong bias.
In GO-Explore this is avoided by performing a random action at the end of a rollout so that the exploration is not conditioned by the current policy. However, the combinatorial generation of goals/labels might be more challenging in this case and thus leading to fewer data per rollout. Ideally, this should be evaluated and compared in the experimental results. However, I have to say that the experimental evaluation is already quite elaborated and provide interesting insights into the performance of the algorithm.

Overall, this is a nice paper with an interesting algorithm, which could be made better by more discussion on some related work and potential limitations.


-----------
Update after rebuttal:
I am very pleased by the answers of the authors, in particular, with the additional experiment showing that the algorithm could be extended with more advanced exploration strategies. I reviewed my rating accordingly.

---

> ### Author Response · Authors · 2020-11-13
> **Response to Reviewer 1**
>
>
> Thank you for your valuable feedback! To address your concerns about the potential lack of exploration, we ran an experiment that shows without any reconfiguration, GCSL can be combined with an exploration strategy to potentially avoid these challenges (**Appendix A.7**, also detailed below).
>
> Thank you for bringing (Ecoffet et al, 2020) to our attention -- we have added it to the related work (Section 4, pg 6). The Go-Explore algorithm is certainly related, but it may not be directly comparable to GCSL. Go-Explore **uses** a goal-reaching algorithm (in policy-based Go-Explore, specifically PPO+SIL) as a sub-procedure to efficiently explore in a single-task sparse reward environment. In principle, GCSL could be used as a drop-in replacement instead of PPO+SIL for the Go-Explore algorithm. This, amongst other ways of using goal-directed policy learning for more general RL tasks, would be a very interesting direction for future work.
>
> GCSL indeed can be susceptible to lack of exploration. This issue is actually a challenge for most self-imitation learning algorithms and policy gradient methods, which tend to local optima without exploring, as discussed in [Oh 17]. Since GCSL can learn with off-policy data, GCSL can deal with these exploration challenges by collecting data using an exploration strategy (e.g. count-based), or by injecting (potentially structured) noise into the data collection process.
>
> To demonstrate this, we combined GCSL with an exploration strategy akin to Go-Explore, now detailed in **Appendix A.7**. During data collection, the greedy policy is followed for the first 90% of the trajectory (getting to the fringe states), and then random actions are taken for the remainder (exploring beyond the fringe). We tested this strategy on a larger version of the four rooms domain, where GCSL with a greedy exploration strategy learns very slowly, but when combined with this new exploration strategy, is able to reach 15% more goals (displayed in Figure 11). Although a full analysis of exploration strategies is beyond the scope of this paper, we hope this result indicates the promise of combining exploration strategies with GCSL to potentially avoid the traditional challenges of self-imitation.
>
> [1] Junhyuk Oh, Yijie Guo, Satinder Singh, and Honglak Lee. Self-imitation learning.

---

> > ### Comment · AnonReviewer1 · 2020-11-23
> > **Review update**
> >
> > Dear authors,
> >
> > Thank you for your response. I have updated my review accordingly.
> >
> > Best regards,

---

### Official Review · AnonReviewer3 · 2020-11-02

**Rating:** 7
**Confidence:** 3

**Review:**

the authors of the paper propose a new way to learn goal-reaching policies by utilizing the previously collected trajectories in an iterative manner. Their novel approach, called goal-conditioned supervised learning (GCSL), learns to reach goals from a target distribution by running the policy and collecting suboptimal trajectories, and then relabeling these collected trajectories during training to perform supervised learning on them to update the policy. This behavioral cloning is done iteratively until convergence.

Relatively ample evaluations and experiments were conducted in multiple settings (5 control environments). GCSL outperforms some RL algorithms in these tasks and is shown to be more robust to hyperparameters.

This work is very close to hindsight relabelling methods [Schaul et al., 2015; Andrychowicz et al., 2017; Rauber et al. 2017], but authors state that their method is more stable and does not estimate a value function.

Pros:
The paper is well written and structured.
The idea of using previous rollouts and relabelling them to use as training data iteratively is very interesting.
The analysis and ablation of learned behaviors with different data collection and relabelling settings is well done.

Cons:
While many ablation studies have been done, the impact of the frequency of performing supervised learning via behavioral cloning on the performance of GCSL is not clear. How many trajectories are collected and relabelled before every policy update via behavioral cloning? and did you do any ablation studies for this?

---

> ### Author Response · Authors · 2020-11-13
> **Response to Reviewer 3**
>
> Thank you for your valuable feedback!
>
> In our hyperparameter robustness analysis, we found that GCSL performs as well when the ratio of policy updates to data collection steps is increased. We have just added a new figure, **Figure 8 in Appendix A.4 (pg 14)** with learning curves when this hyperparameter is ablated. By default, we perform one gradient update for every environment step that is collected -- in our ablation, we find that performing policy update steps 2x and 4x as often leads to similar final performance. Please let us know if this addresses your concern!

---

### Author Response · Authors · 2020-11-13
**Updated manuscript with new experiments, analyses, and clarifications**


We thank the reviewers for their positive assessment of our work and helpful suggestions for improvement. Please find responses to specific questions directly commented after each review.

We have updated the manuscript and included related work, clarified experimental details about the policy architecture and RL comparisons, and added a discussion about the relevance of the final timestep objective. Based on suggestions from reviewers, we have also added a number of additional experimental results to the paper.

### Additional experiments and plots:

_Alternative metrics:_ Upon suggestions from Reviewers 2 and 4, we have provided alternate metrics for evaluating the goal conditioned policies (“success at final timestep”, “time to reach goal”, “minimum distance to goal”, and “success at any timestep”) in Figure 12.

_Frequency of Policy Updates:_ As suggested by Reviewer 3, we have added a new plot showing that GCSL remains performant when the frequency of policy updates is increased (Figure 8).

_Exploration strategies:_ As suggested by Reviewer 1, we provide a demonstration that GCSL can avoid issues of exploration by using exploration strategies. We evaluate GCSL combined with a simple exploration strategy inspired by Go-Explore, enabling GCSL to solve a larger environment requiring more directed exploration (Appendix A.7, Fig 11).

_Quantitative metric of “directness”:_  As suggested by Reviewer 2, we provide a quantitative metric for how quickly the goal is reached by GCSL policies (Appendix C.1 Table 1).

---

### Decision · Program_Chairs · 2021-01-07
**Final Decision**

**Decision:**

Accept (Oral)

**Comment:**

The paper leverages concepts coming from hindsight relabelling methods to define a novel "iterated" supervised learning procedure to learn policies to reach different goals. The algorithmic solution is well supported in terms of intuition, preliminary theoretical guarantees, as well as strong empirical validation.

There is a general consensus among the reviewers that this is a strong submission and the rebuttal helped in clarifying some aspects of the paper (e.g., the comparison with Go-Explore) and reinforced the empirical analysis. This is a clear accept.